# Palmitic Acid Promotes Lung Metastasis of Melanomas via the TLR4/TRIF-Peli1-pNF-κB Pathway

**DOI:** 10.3390/metabo12111132

**Published:** 2022-11-17

**Authors:** Xuedan Zhang, Xiaoyu Li, Guohang Xiong, Fang Yun, Yu Feng, Qinxuan Ni, Na Wu, Lijuan Yang, Zihan Yi, Qiao Zhang, Zhe Yang, Yingmin Kuang, Buqing Sai, Yuechun Zhu

**Affiliations:** 1Department of Biochemistry and Molecular Biology, School of Basic Medicine, Kunming Medical University, Kunming 650500, China; 2Department of Medical Oncology, The Third Affiliated Hospital of Kunming Medical University (Tumor Hospital of Yunnan Province), Kunming 650500, China; 3Department of Pathology, The First Affiliated Hospital of Kunming Medical University, Kunming 650500, China; 4Department of Organ Transplantation, The First Affiliated Hospital of Kunming Medical University, Kunming 650500, China

**Keywords:** melanoma, palmitic acid, TLR4/TRIF-Peli1-pNF-κB axis

## Abstract

A high-fat diet plays an important role in aggravating cancers. Palmitic acid (PA) is one of the components of saturated fatty acids; it has been reported to promote tumor proliferation in melanomas, but the signal transduction pathway mediated by palmitic acid remains unclear. This study showed that palmitic acid can promote the lung metastasis of melanomas. Moreover, the interaction between palmitic acid and toll-like receptor 4 (TLR4) was predicted by molecular docking. The experimental results proved that palmitic acid could promote the TLR4 and Toll/IL-1 receptor domain-containing adaptor-inducing IFN-β (TRIF) expression. The expression of Pellino1 (Peli1) and the phosphorylation of NF-kappa B (pNF-κB) were downregulated after the suppression of TLR4 and the silencing of Peli1 also inhibited the phosphorylation of NF-κB. Therefore, we concluded that palmitic acid promoted the lung metastasis of melanomas through the TLR4/TRIF-Peli1-pNF-κB pathway.

## 1. Introduction

Melanomas originate from melanocytes in the skin and other organs, and are characterized by a high malignancy, a rapid progression, an early metastasis, insensitivity to radiotherapy and chemotherapy, and a high mortality [1]. There is increasing evidence that a high-fat diet can promote the occurrence and development of tumors [2,3]. However, the specific component of a high-fat diet that is responsible for oncogenesis remains unknown. Palmitic acid (PA) is a 16 carbon long-chain saturated fatty acid; it is a common saturated fatty acid in many common dietary fats, accounting for about 13% of the total fatty acid content of peanut oil, about 65% of butter, about 42% of lard, and about 53% of tallow. Studies have shown that PA can promote the occurrence and development of many diseases such as tumors, cardiovascular disease, and inflammation [4]. However, the effect of PA on melanomas remains unclear.

Toll-like receptor 4 (TLR4), a member of the TLR-like families, can activate TRIF pathways [5]. The toll/IL-1 receptor domain-containing adaptor-inducing IFN-β (TRIF) is a unique adapter of the TLR3 and TLR4-mediated signaling pathways. It can activate TLR4-mediated signal transduction, thus regulating interferon regulatory factor 3 (IRF3) and activating the phosphorylation of NF-kappa B (pNF-κB) expression, thereby promoting the expression of inflammatory factors [6]. TLR4/NF-κB can promote the proliferation and invasion of melanomas [7]. In addition, PA can promote the proliferation of colorectal cancer by promoting the expression of TLR4 [8]. However, there is no relevant report on melanomas. Pellino1 (Peli1) is an E3 ubiquitin ligase, which is essential for regulating TLR signals [9]. Peli1 knockout significantly inhibits NF-κB activation and pro-inflammatory gene expression [10]. It has been reported that the accumulation of Peli1 can promote the proliferation and migration of thyroid papillary carcinomas [11]. In esophageal squamous cell carcinomas, Peli1 improved the radiotherapy sensitivity and promoted tumor cell apoptosis [12]. However, the function and related signal transduction of Peli1 in melanomas are not clear. Given the role of Peli1 in other tumors and that of TLR4/NF-κB in melanomas, this study examined the effect of Peli1 and the related signaling pathways that influence the occurrence and development of melanomas.

## 2. Materials and Methods

### 2.1. Drugs

For the in vitro experiments, PA (P9767, Sigma-Aldrich, St. Louis, MO, USA) was dissolved in 10% BSA (A8850, Solarbio Life Sciences, Beijing, China) with gentle warming, as previously described [13]; the concentration of the stock solution was 2 mM. TAK-242 (HY-11100, Medchemexpress, Princeton, NJ, USA) was dissolved in dimethyl sulfoxide (DMSO) (D8371, Solarbio Life Sciences, Beijing, China); the concentration of the stock solution was 10 mM.

### 2.2. Animals and Treatment

A total of 20 C57BL/6J mice (6 weeks old, male) were purchased from the Experimental Animal Center of Kunming Medical University (License number: K2020-0006). The mice were housed in a room at 22 ± 2 °C with a 12 h of light /12 h of dark cycle and allowed free access to food and water. All animal procedures were conducted according to the National Institutes of Health (NIH) Guide for the Care and Use of Laboratory Animals. After 1 week of acclimation, the mice were randomly divided into two groups (*n* = 10). The mice of the PA group were fed with a high-PA diet (10% PA) for two months whereas the mice of the control group were fed with a normal diet for two months. The serum cholesterol of the mice was used as the marker of hyperlipidemia. The mice were then intravenously injected with B16 cells (1 × 10^6^/mL) in a volume of 100 μL, as in a previous study [14]. After tumor growth for 15 days, the mice were anesthetized and dissected. The lungs of the mice were extracted. The lungs were preserved in 4% paraformaldehyde at 4 °C. After two weeks of fixation, the lungs were immersed in increasing concentrations of sucrose (10%, 20%, and 30%) and sectioned (5 μm) on a freezing microtome (CM 1860, Leica Biosystems, Germany, Weztlar) at −20 °C. For the detection of proteins, the tumor in the lung was extracted on ice and stored at −80 °C.

### 2.3. Cell Culture and Treatment

A375 cells and B16 cells (Cell Bank of the Chinese Academy of Science, Shanghai, China) were cultured in a high-glucose medium (06-105-57-1ACS, Biological Industries, Israel) containing 10% fetal bovine serum (04-001-1ACS, Biological Industries, kibbutz beit-haemek, Israel) at 37 °C, 90% humidity, and 5% CO_2_. For the cell proliferation detection, the cells were treated with 12.5, 25, or 50 μM PA for 24 h or treated with 0.1, 1, 10, or 50 μM TAK-242 (a TLR4 antagonist). In addition, 50 μM TAK-242 was used as a pretreatment for 1 h before the addition of PA. The concentrations were based on previous studies with slight modifications [15,16].

### 2.4. Cell Proliferation Detection

The cells were seeded at a density of 5 × 10^3^ cells/well in a 96-well plate. After the cells were incubated with drugs following the above procedures, the cell proliferation was detected using an MTS kit (CTB169, Promega, WI, USA). The absorbance at 490 nm was read on a microplate reader (51119200, Thermo Fisher Scientific, Waltham, MA, USA) and the relative cell proliferation was calculated according to the formula provided in the kit.

### 2.5. siRNA Transfection

The transient transfection of siRNA (12792, GenePharma, Shanghai, China) was performed using a Lipofectamine 2000 transfection reagent (11668500, Thermo Fisher Scientific, Waltham, MA, USA) according to the manufacturer’s instructions. Briefly, the melanoma cells were seeded in 6-well plates and transfected with 100 pmol/well siRNA for 24 h using 5 μL Lipofectamine 2000. The siRNA targeting sequences of Peli1 were as follows. Set 1: 5′-GGUGGUUGAAUAAUACUCAUTT-3′, 5′-AUGAGUAUAUUCAACCACCTT-3′; Set 2: 5′-GUCAGUACAAAGCACUAUATT-3′, 5′-UAUAGUGCUUUGUACUGACTT-3′; and Set 3: 5′-CAGCAUAGCAUAUCAUAUATT-3′, UAUAUGAUAUGCUAUGCUGTT-3′. The sequence of the negative control siRNA was 5′-UUCUUCGAACGUGUCACGUTT-3′, 5′-ACGUGACACGUUCGGAGAATT-3′.

### 2.6. Hematoxylin–Eosin (HE) Staining

The lung sections were stained using an HE staining kit (C0105S, Beyotime, Shanghai, China). The lung sections were stained with the hematoxylin solution for 8 min, followed by rinsing with distilled water. The lung sections were then stained with the eosin solution for 1 min, followed by immersing in increasing concentrations of ethanol (70%, 80%, 90%, and 100%). Finally, the lung sections were cleared with xylene for 5 min. The images of the HE staining were obtained using a slice scanner (KF-Pro-005, KFBio, Shanghai, China).

### 2.7. Immunofluorescence

The cells and tissue sections were fixed with 4% paraformaldehyde for 30 min, washed with 0.1% sodium citrate (C-1032, Solarbio Life Sciences, Beijing, China) three times, perforated with 0.1%Triton X-100 (T-8200, Solarbio Life Sciences, Beijing, China) for 30 min, sealed with 5% goat serum at room temperature for 2 h, and incubated with the primary antibody overnight. The primary antibody was washed off with 0.1% PBST three times and the section was incubated with the secondary antibody at 37 °C for 2 h and then finally stained with DAPI. The main antibodies were as follows: anti-TLR4 (66350-1-Ig, Proteintech, Chicago, IL, USA); anti-TRIF (23288-1-AP, Proteintech, Chicago, IL, USA); Peli1 (199336, Abcam, Cambridge, UK); pNF-κBp65 (3036, Cell Signaling Technology, MA, USA); CoraLite488-conjugated Affinipure goat anti-mouse IgG (H+L) (SA00013-1, Proteintech, Chicago, IL, USA); and Cy3-conjugated Affinipure goat anti-rabbit IgG (H+L) (SA00009-2, Proteintech, Chicago, IL, USA).

### 2.8. Western Blot Analysis

After the treatment, the cells were washed with pre-cooled PBS and then lysed with a RIPA lysis buffer (R0020, Solarbio Life Sciences, Beijing, China) containing protease and phosphatase (1:100) inhibitors (P0100, Solarbio Life Sciences, Beijing, China) on ice for 30 min. The lysate was centrifuged at 12,000 rpm for 10 min at 4 °C. The protein concentration was detected using a BCA^TM^ protein assay kit (P1511, Applygen, Beijing, China). The protein in the lysate was analyzed by 8–15% SDS-PAGE and transferred to a PVDF (IPVH00010, Millipore, Phillipsburg, NJ, USA) membrane. The membrane was blocked with 5% skimmed milk at 37 °C for 1 h. The membrane was then incubated with the primary antibody overnight at 4 °C. The next day, the membrane was washed and incubated with the HRP-bound secondary antibody for 1 h at room temperature. Finally, the chemiluminescent signals were amplified using a chemiluminescence reagent ECL kit (K-12045-D50, Bioship, Shanghai, Beijing) and a high-quality image was acquired using a Bio-Rad ChemiDoc XRS system (Bio-Rad, Hercules, CA, USA). β-actin was used as the reference control. All experiments were repeated at least three times. The main antibodies were as follows: anti-TLR4 (66350-1-Ig, Proteintech, Chicago, IL, USA); anti-TRIF (23288-1-AP, Proteintech, Chicago, IL, USA); Peli1 (199336, Abcam, Cambridge, UK); pNF-κBp65 (3036, Cell Signaling Technology, Danvers, MA, USA); MMP2 (10373-2-AP, Proteintech, Chicago, IL, USA); E-cadherin (20874-1-AP, Proteintech, Chicago, IL, USA); Vimentin (10366-1-AP, Proteintech, Chicago, IL, USA); and anti-β-actin (66009-1-Ig, Proteintech, Chicago, IL, USA).

### 2.9. Molecular Docking

Molecular docking was performed by autodock v4.2.6 and autodocktools v1.5.6, based on previous studies [17]. Human TLR4 structure and ligand information were obtained from RCSB PDB “https://pubchem.ncbi.nlm.nih.gov/ (accessed on 15 April 2022)”. First, the acceptor and ligand were prepared using autodocktools v1.5.6, including removing the water molecules, adding hydrogen, and calculating the number of atoms. The docking pocket of PA and TLR4 referred to the binding ligand position in the TLR4 crystal structure. PA and TLR4 were then docked using autodock v4.2.6. In addition, the ligand bound in the TLR4 crystal structure was removed and then spliced back to the original position, which generated a docking score as a criterion for evaluating good docking. Finally, the binding sites and interactions between PA and TLR4 were calculated using a protein–ligand interaction profiler “https://plip-tool.biotec.tu-dresden.de/plip-web/plip/index (accessed on 15 April 2022)”. The visualization and analysis were performed by PyMOL v2.5.2.

### 2.10. Statistical Analysis

All data were analyzed by SPSS v26.0 software and presented as the mean ± standard deviation (SD). An independent samples *t*-test was used to analyze the data from Figure 1 (except Figure 1C), 3 and Appendix A. A one-way analysis of variance (ANOVA) was used to analyze the data from Figure 1C, Figure 4A, and Appendix A. An independent samples *t*-test was used to analyze the data from Figure 6A. two-way ANOVA (no repeated measures) was used to analyze the data from the other experiments. The factors were defined as PA and TAK-242 treatments, and PA and si-Peli1 treatments, respectively. Before the ANOVA, the normality and homogeneity of equal variance were confirmed. The level of significance was set at 0.05.

## 3. Results

### 3.1. PA Promotes Melanoma Cell Proliferation, Invasion, and Lung Metastasis

First, we investigated the cholesterol level in the serum of mice after a high-PA diet for two months. The high-PA diet group had higher cholesterol levels compared with the normal diet group (Appendix A). The mice then received a tail-vein injection of melanoma cells (B16 cells), followed by a tumor growth period for 15 days. There were more metastases in the lungs of the mice in the high-PA diet group (Figure 1A). The epithelial–mesenchymal transition (EMT) plays an important role in tumor invasion and metastasis [18]. The EMT is characterized by changes in the morphology and molecular markers. Cells acquire the mesenchymal phenotype from a loss of epithelial-related markers and mesenchymal-related markers (including N-cadherin and Vimentin) and matrix metalloproteinases (MMPs) are upregulated [19]. Therefore, we detected E-cadherin, MMP2, and Vimentin expressions. The results showed a lower E-cadherin expression and higher MMP2 and Vimentin expressions in the tumors of the high-PA diet group (Figure 1B). These results indicated that PA promoted the lung metastasis of the melanomas. In addition, PA (25 μM) significantly promoted cell proliferation and invasion in B16 and A375 cells (Figure 1C,D), with a lower E-cadherin expression and higher MMP2 and Vimentin expressions compared with the control group (Figure 1E,F).

### 3.2. Analysis of Protein-Binding Model of PA and TLR4

To predict the potential molecules that interacted with PA, the molecular docking results showed that there were two binding pockets between the original ligand and TLR4, which were located on the C-chain and D-chain of TLR4 (Figure 2). Among them, the docking score of the proto-ligand and TLR4 was −5.5 kcal/mol on the C-chain and the docking score of the proto-ligand and TLR4 was −5.6 kcal/mol on the D-chain. Interestingly, the docking score of PA and TLR4 was −6.0 kcal/mol on the C-chain docking pocket and the docking score of PA and TLR4 was −6.4 kcal/mol on the D-chain docking pocket. Irrespective of the C-chain or D-chain, the docking reliability of PA and TLR4 was higher than the verification standard, which suggested that the docking result was reliable. These results indicated that there was a good binding site between PA and TLR4.

### 3.3. PA Can Activate the TLR4/TRIF-Peli1-pNF-κB Pathway

PA significantly activated the TLR4/TRIF-Peli1-pNF-κB pathway (Figure 3). In the metastasis sites, the high-PA diet group fluorescence intensity was stronger and the protein expression level was higher (Figure 3A,B). PA (25 μM) also activated the TLR4/TRIF-Peli1-pNF-κB pathway in the A375 cells (Figure 3C) and B16 cells (Figure 3D). These results suggested that PA promoted the metastasis of melanomas via the TLR4/TRIF-Peli1-pNF-κB pathway.

### 3.4. Inhibition of TLR4 Blocks PA-Induced Proliferation and Invasion in Melanoma Cells

To examine the role of TLR4 in PA-induced proliferation and invasion in melanoma cells, we inhibited the functional activity of TLR4 using a TLR4 selective antagonist (TAK-242). The results showed that TAK-242 inhibited the proliferation of B16 cells and A375 cells in a dose-dependent manner (Figure 4A). We selected TAK-242 (50 μM) for the subsequent experiments. Interestingly, there was a higher E-cadherin expression and lower MMP2 and Vimentin expressions compared with the control group when TAK-242 (50 μM) was administered alone in the B16 cells (Figure 4C) and A375 cells (Figure 4D). TLR4 seemed to play an important role in the proliferation and invasion of the melanoma cells. Furthermore, a TAK-242 (50 μM) pretreatment strongly blocked the promoting effect of PA on the proliferation of the B16 cells and A375 cells (Figure 4B). The TAK-242 (50 μM) pretreatment also attenuated a PA-induced low expression of E-cadherin and high expressions of MMP2 and Vimentin in B16 cells (Figure 4C) and A375 cells (Figure 4D). These results suggested that the cell proliferation and invasion induced by PA were mediated by TLR4. Moreover, the TAK-242 (50 μM) pretreatment also blocked the activation of the TLR4/TRIF-Peli1-pNF-κB pathway induced by PA in the A375 cells (Figure 5A,B) and B16 cells (Figure 5C,D). Compared with the PA group, the protein expression level of TAK-242 with the PA administration was significantly reduced whereas the activation level of this pathway was lower compared with the control group when TAK-242 (50 μM) was administered alone. These results indicated that PA promoted the proliferation and invasion of melanoma cells through TLR4 signaling.

**Figure 4 metabolites-12-01132-f004:**
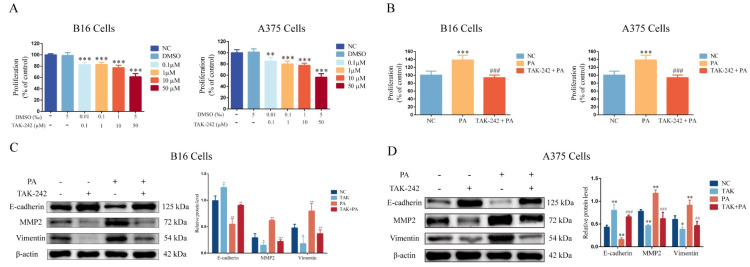
TAK-242 treatment inhibited the proliferation, and changed the expression, of EMT related proteins induced by PA in the B16 and A375 cells. (**A**) The level of cell proliferation after B16 and A375 cells were incubated with TAK-242. (**B**) The level of cell proliferation after B16 and A375 cells were incubated with TAK-242 and/or PA for 24 h. (**C**) Protein levels and quantitative analysis of E-cadherin, MMP2, and Vimentin in B16 cells incubated with TAK-242 and/or PA for 24 h. (**D**) Protein levels and quantitative analysis of E-cadherin, MMP2, and Vimentin in A375 cells incubated with TAK-242 and/or PA for 24h. In A375 and B16 cells, treatment with PA or TAK-242 (+) and no treatment with PA or TAK-242 (-). ** p* < 0.05, *** p* < 0.01, or **** p* < 0.001 compared with the control group. *## p* < 0.01, or *### p* < 0.001 compared with the PA group; the significance of the ANOVA.

### 3.5. Inhibition of Peli1 Expression Alleviates the Impact of PA on the Proliferation and Invasion of Melanoma Cells and Peli1-PNF-κB Signaling

The expression of Peli1 in the B16 cells and A375 cells was inhibited using an siRNA-targeting Peli1 pretreatment (Appendix A). The results showed that the si-Peli1 pretreatment for 24 h, 48 h, or 72 h significantly restrained the proliferation of the B16 cells and A375 cells (Figure 6A). In addition, there was a higher E-cadherin expression and lower MMP2 and Vimentin expressions compared with the control group when si-Peli1 was administered alone in the B16 cells (Figure 6C) and A375 cells (Figure 6D). Furthermore, the inhibition of the Peli1 expression blocked PA-induced cell proliferation and changes in the E-cadherin, MMP2, and Vimentin expressions in the B16 cells and A375 cells (Figure 6B–D). The inhibition of the Peli1 expression also blocked the activation of Peli1-pNF-κB signaling induced by PA in the A375 cells (Figure 7A–D) and B16 cells (Figure 7E), but had no effect on TLR4/TRIF signaling (Figure 7). In addition, the activation level of Peli1-pNF-κB signaling was lower compared with the control group when si-Peli1 was administered alone (Figure 7). These results suggested that Peli1, as a downstream signal of TLR4, played an important role in PA-induced melanoma cell proliferation and invasion.

**Figure 6 metabolites-12-01132-f006:**
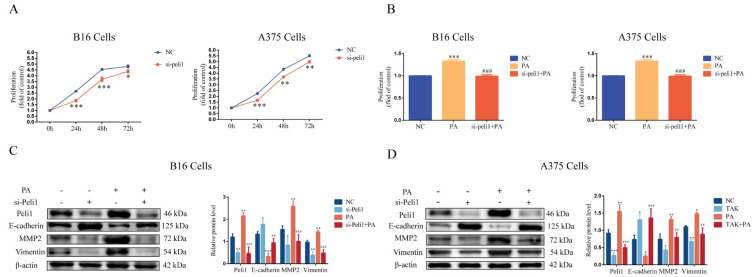
Inhibition of Peli1 expression blocked PA-induced cell proliferation and changed the expression of EMT-related proteins in the B16 and A375 cells. (**A**) The level of cell proliferation after si-Peli1 in B16 and A375 cells. (**B**) The level of cell proliferation after B16 and A375 cells were incubated with si-Peli1 and/or PA for 24 h. (**C**) Protein levels and quantitative analysis of Peli1, E-cadherin, MMP2, and Vimentin in B16 cells incubated with si-Peli1 and/or PA for 24 h. (**D**) Protein levels and quantitative analysis of Peli1, E-cadherin, MMP2, and Vimentin in A375 cells incubated with si-Peli1 and/or PA for 24 h. In A375 and B16 cells, treatment with PA or si-Peli1 (+) and no treatment with PA or si-Peli1 (-). ** p* < 0.05, *** p* < 0.01, or ****p* < 0.001 compared with the control group. *## p* < 0.01, or *### p* < 0.001 compared with the PA group; the significance of the ANOVA.

## 4. Discussion

The results of this study suggested that PA plays an important role in melanoma development. At the same time, based on molecular docking and subsequent animal experiments, we hypothesized that PA may be a ligand of TLR4. Its role in mediating the TLR4/TRIF-Peli1-pNF-κB axis pathway may provide a new idea for the treatment of melanomas.

Two of the characteristics of aggressive tumor cells are their ability to migrate and invade. The phenotype of these cells undergoes a transformation from epithelial cells to mesenchymal cells. This process is called EMT [20]. The downregulation of E-cadherin is the basis of EMT [21]. MMPs are important in regulating cell migration and invasion by digesting the extracellular matrix (ECM) [22]. More importantly, the expression of Vimentin is upregulated and the cell motility is enhanced [23]. EMT is the basis of metastasis [24]. After the PA treatment, the lung metastasis of the mice was significantly increased and the corresponding expression of MMP2 and Vimentin was upregulated whereas the expression of E-cadherin was downregulated. These results indicate the importance of PA in melanoma metastasis.

TLR4 plays an important role in melanoma development. Previous studies have shown that TLR4 can protect cells from TNFα-induced apoptotic damage. In addition, knocking out TLR4 can significantly reduce melanoma metastasis [7]. After silencing TLR4, the cell migration ability of melanomas was decreased [25]. TLR4 increases STAT3 activation and promotes angiogenesis and EMT [26]. TLR4 activates NF-κB to promote the occurrence and development of melanomas [27]. In addition, the UV-induced activation of TLR4 promotes the inflammatory production of neutrophils as well as the angiogenesis and metastasis of melanomas [28]. In this study, we found that TLR4 could promote cell proliferation and invasion by activating TRIF-Peli1-pNF-κB. The activation of pNF-κB can promote EMT [29,30], but the specific mechanism remains unclear. Based on our results, we plan to study whether NF-κB can target the promoter region of EMT-related factors and regulate their transcription to promote melanoma metastasis.

Increasing evidence suggests that a high-fat diet is strongly linked to cancer. In a few cancers, obesity and a high fat intake increase the risk of cancer occurrence and progression [31,32]. PA is a saturated fatty acid, which has adverse effects on many chronic diseases [4,33] and promotes tumor proliferation [2,16]. We speculated that PA may promote tumor proliferation as a molecular signal of tumor cells. In melanomas, PA can activate the Akt (also known as protein kinase B, or PKB) pathway to promote melanoma proliferation [16]. In pancreatic cancers, PA activates the TLR4/ROS/NF-κB/MMP9 signaling pathway, increasing cancer aggressiveness [34]. In rectal cancer, PA promotes TLR4 expression and tumor proliferation by promoting the expression of the transcription factor PU.1 upstream of TLR4 [8]. However, in the present study, we indicated that TLR4 was a ligand of PA, which promoted the lung metastasis of melanomas by activating TLR4 as well as the nuclear translocation of pNF-κB, influencing the translation of E-cadherin, MMP2, and Vimentin. However, the relationship between pNF-κB and E-cadherin, MMP2, and Vimentin needs further study.

## 5. Conclusions

In summary, the results of this study suggested that TLR4 may be a ligand of PA, promoting the lung metastasis of melanomas by triggering downstream TRIF-Peli1-pNF-κB to induce cell migration and invasion (Figure 8).

## Figures and Tables

**Figure 1 metabolites-12-01132-f001:**
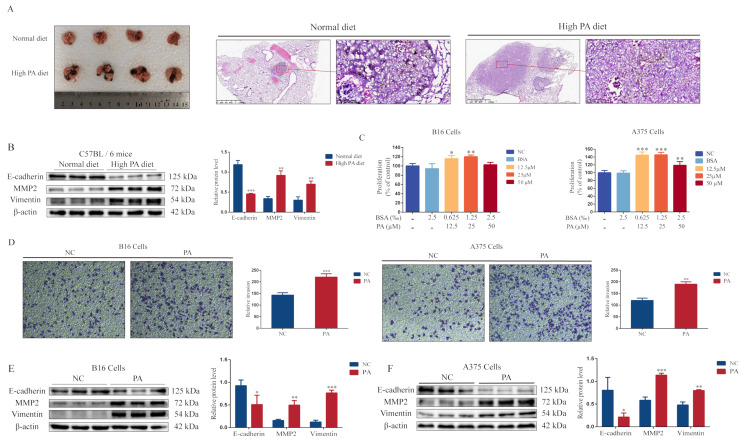
Palmitic acid promotes lung metastasis of melanomas. (**A**) Overall lung changes and HE staining after animal modeling (representative images). (**B**) The protein blots of E-cadherin, MMP2, Vimentin, and quantitative analysis from a mouse model. (**C**) The level of cell proliferation after B16 and A375 cells were incubated with PA for 24 h at different concentrations (12.5, 25, and 50 μM). (**D**) The invasiveness of B16 and A375 cells. (**E**) The protein blots of E-cadherin, MMP2, Vimentin, and quantitative analysis from B16 cells. (**F**) The protein blots of E-cadherin, MMP2, Vimentin, and quantitative analysis from A375 cells. ** p* < 0.05, *** p* < 0.01 or **** p* < 0.001 compared with the control group; the significance of the *t*-test.

**Figure 2 metabolites-12-01132-f002:**
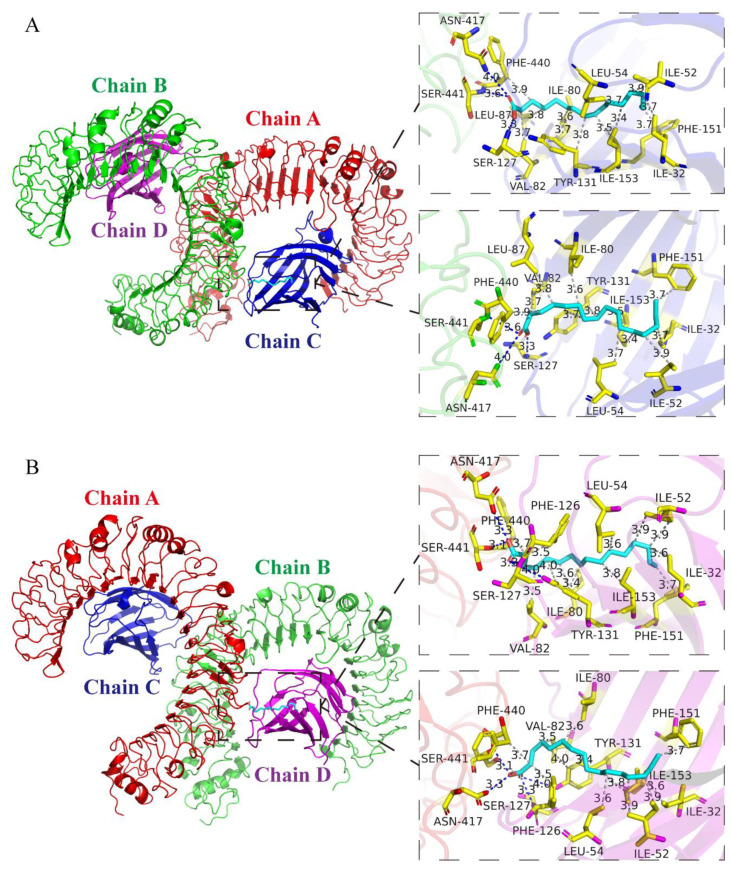
Molecular docking between PA and TLR4 Four chains of TLR4 are shown as multicolor bands. PA is shown as cyan sticks and TLR4 residues are marked with yellow sticks. (**A**) The image shows the docking pose, binding sites, and interactions between PA and the C-chain of TLR4. There were three hydrogen bonds between PA and the TLR4 protein residues, which were Ser-127 at 3.3 Å, Asn-417 at 3.6 Å, and Ser-441 at 3.6 Å. In addition, PA and TLR4 also formed 12 hydrophobic interactions in the C-chain, which were Ile-32 at 3.75 Å, Ile-52 at 3.93 Å, Leu-54 at 3.72 Å, Ile-80 at 3.60 Å, Val-82 at 3.73 Å, Leu-87 at 3.77 Å, Tyr-131 at 3.81 Å, Tyr-131 at 3.73 Å, Phe-151 at 3.66 Å, Ile-153 at 3.36 Å, Ile-153 at 3.54 Å, and Phe-440 at 3.88 Å. (**B**) Docking pose, binding sites, and interactions between PA and D-chain of TLR4. PA formed four hydrogen bonds with the TLR4 protein residues in the binding pocket of the D-chain, which were Ser-127 at 2.33 Å, Tyr-131 at 3.97 Å, Asn-417 at 3.00 Å, and Ser-441 at 2.25 Å. PA and the TLR4 protein residues formed 12 hydrophobic interactions in the binding pocket of the D-chain, which were Ile-32 at 3.57 Å, Ile-52 at 3.87 Å, Ile-52 at 3.94 Å, Ieu-54 at 3.57 Å, Ile-80 at 3.58 Å, Val-82 at 3.53 Å, Phe-126 at 3.5 Å, Tyr-131 at 3.4 Å, Tyr-131 at 3.95 Å, Phe-151 at 3.7 Å, Ile-153 at 3.8 Å, and Phe-440 at 3.73 Å.

**Figure 3 metabolites-12-01132-f003:**
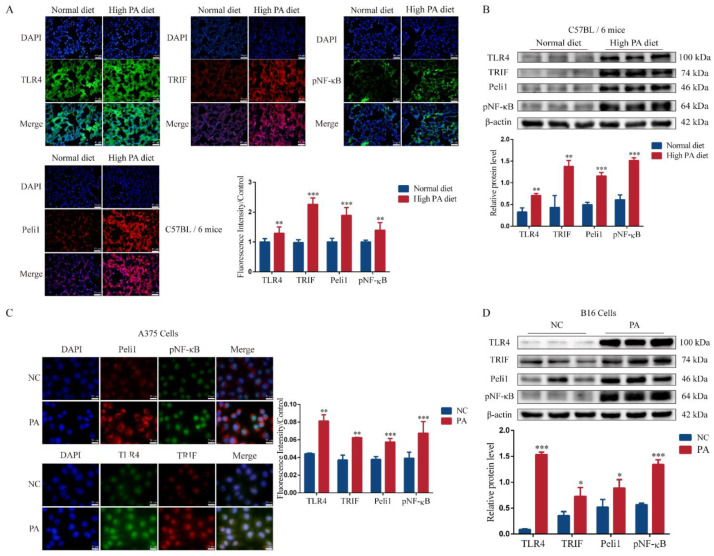
PA activated the TLR4/TRIF-Peli1-pNF-κB pathway in melanomas. (**A**) The fluorescence intensity and quantitative analysis of TLR4, TRIF, Peli1, and pNF-κB in mice lung tissues. (**B**) The blots of protein and quantitative analysis of mice lung tissues. (**C**) The fluorescence intensity and quantitative analysis of TLR4, TRIF, Peli1, and pNF-κB in A375 cells. (**D**) The blots of protein and quantitative analysis of B16 after PA treatment for 24 h. ** p* < 0.05, *** p* < 0.01, or **** p* < 0.001 compared with the control group.

**Figure 5 metabolites-12-01132-f005:**
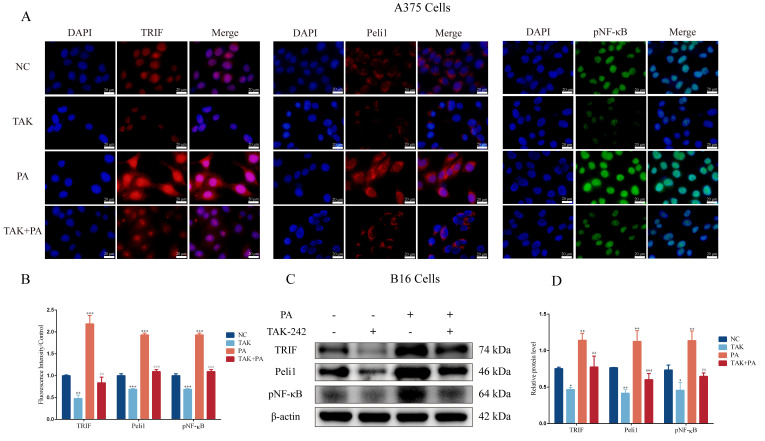
The fluorescence intensity and protein expression of TLR4/TRIF-Peli1-pNF-κB pathway after TAK-242 treatment. (**A**) The fluorescence intensity of TRIF, Peli1, and pNF-κB in A375 cells. (**B**) Quantitative analysis of fluorescence intensity. (**C**) The blots of TRIF, Peli1, and pNF-κB in B16 cells. (**D**) Quantitative analysis of TRIF, Peli1, and pNF-κB expression in B16 cells. In A375 and B16 cells, treatment with PA or TAK-242 (+) and no treatment with PA or TAK-242 (-). ** p* < 0.05, *** p* < 0.01, or **** p* < 0.001 compared with the control group. *## p* < 0.01 or *### p* < 0.001 compared with the PA group; the significance of the ANOVA.

**Figure 7 metabolites-12-01132-f007:**
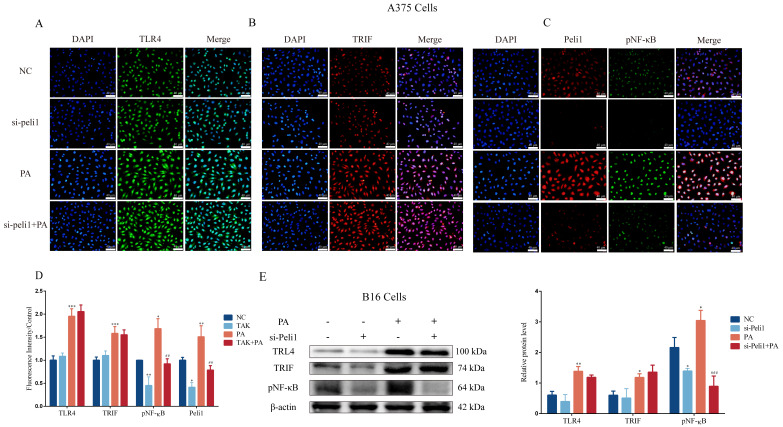
The fluorescence intensity and protein expression of TLR4/TRIF-Peli1-NF-κB pathway after si-Peli1. (**A**) The fluorescence intensity of TLR4 in A375 cells. (**B**) The fluorescence intensity of TRIF in A375 cells. (**C**) The fluorescence intensity of Peli1 and pNF-κB in A375 cells. (**D**) Quantitative analysis of fluorescence intensity. (**E**) The protein blots and quantitative analysis of TLR4, TRIF, and pNF-κB in B16 cells. In A375 and B16 cells, treatment with PA or si-Peli1 (+) and no treatment with PA or si-Peli1 (-). ** p* < 0.05, *** p* < 0.01, or **** p* < 0.001 compared with the control group. *## p* < 0.01 or *### p* < 0.001 compared with the PA group; the significance of the ANOVA.

**Figure 8 metabolites-12-01132-f008:**
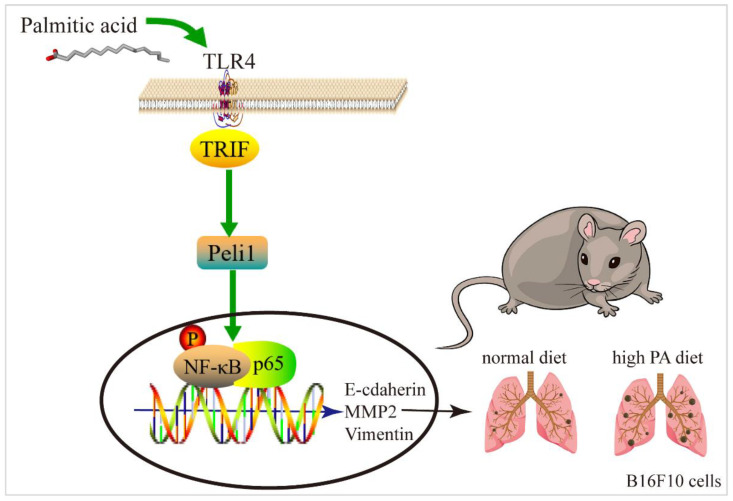
Mechanism simulation diagram of melanoma lung metastasis.

## Data Availability

The original contributions presented in the study are included in the article/supplementary material, further inquiries can be directed to the corresponding authors.

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
