# Peer review of "Palmitic Acid Promotes Lung Metastasis of Melanomas via the TLR4/TRIF-Peli1-pNF-κB Pathway"

_metabolites, 2022, doi:10.3390/metabo12111132_

Round 1

Reviewer 1 Report

Zhang et al., reported “Palmitic acid promotes lung metastasis of melanoma via TLR4/TRIF-Peli1-NF-κB pathway” is complete and well within the scope of this study. This study reports Palmitic acid is promoting promote tumor proliferation in melanoma, but the signal transduction pathway mediated by palmitic acid remains unclear. In addition, the interaction between palmitic acid and Toll-like receptor 4 (TLR4) was predicted by molecular docking. This study revealed that palmitic acid can promote TLR4 and Toll/IL-1 receptor domain containing adaptor inducing IFN-β (TRIF) expression. This study concluded that palmitic acid promoted lung metastasis of melanoma through TLR4/TRIF-Peli1-NF-κB pathway. This MS is well-organized, well-written, and of interest to the reader. However, this study has some minor issues that need to be addressed before acceptance.

Comments

1. So many typographical errors throughout the manuscript.

2. The findings of the study were poorly interpreted. It should be improved with relevant references.

3. Authors must include conclusion of the study findings.

Author Response

Dear Reviewer,

Thank you for your comments on our manuscript entitled “Palmitic acid promotes lung metastasis of melanoma via TLR4/TRIF-Peli1-NF-κB pathway” (Manuscript Number: 2002846). These comments are completely valuable and helpful for improving our manuscript, as well as the significantly guidance to our research. We appreciated your thoughtful review and have attempted to adequately respond to your concern and suggestion. Revised sections are marked in red in the manuscript. The main revisions and our responses to your comments are as an attachment.

Reviewer 2 Report

The authors in the paper titled:  Palmitic acid promotes lung metastasis of melanoma via TLR4/TRIF-Peli1-NF-κB pathway introduced us to a mechanism of high fat-induced cancer metastases. The paper can be published in Metabolites after major revisions.

Here are the things that should be revised:

1.       Amino acid three-letter abbreviation should be written with cap first letter and small letters. For exp. Phe not PHE

2.       Line 171 “t test” should be t-test

3.       Fig2.

Line 179 Tyr-3.81 at 3.81 Å, should be Tyr-131 at 3.81 Å; and further Tyr-131 at 3.73 Å; Phe-151 at 3.66 Å; Ile-153 at 3.36 Å; Ile-153 at 3.36 Å; Ile-153 at 3.54 Å

Line 180 Phe-440 at 3.88 Å

Line 179 Leu-87 at 3.77 not Leu 878

Line 184 IEU-54 should be Leu-54

Do Tyr and Ile have two hydrophobic connections?

4.       Do the chains C and have the same sequence? Do we have the same amino acids at the same place in both chains? For exp. Tyr-131, Ser-127….

5.       Vimentin should be uniformly written with cap first letter or a small.

6.       Line 223 (Results) “… had hinger…” should be … had higher…

7.       Line 289 “… time, Based…” should be … time, based…

8.       Line 293 EMT abbreviation does not have a description

9.       Line 313 Akt pathway  Akt needs a description

10.   Line 317 Instead of “However, the present study…” it should be,  However, in the present study…

Author Response

(The authors gave the same response as above.)

Reviewer 3 Report

Reviewers’ Comments (Remarks to the Authors):

The manuscript emphasizes into the theory that high-calorie and high-fat diets are considered a high risk factor in tumor progression and consequently in the worse prognosis of patients.

Specifically, the authors describe the importance of palmitic acid PA in promoting the signaling pathway that would lead to greater malignancy of the tumor. The experimental results proved that palmitic acid can promote TLR4/TRIF expression. They demonstrated and concluded that palmitic acid promoted lung metastasis of melanoma through TLR4/TRIF-Peli1-NF-κB pathway

   The authors exemplify the existing crosstalk between several cell types, pathways and their interconnection with the tumor microenvironment (TME). In a detailed way and through a clear grammar, the authors encourage a reading by the reader, making the manuscript attractive, and with a great scientific value; being necessary characteristics to attract readers to the MDPI group. However, some questions must be clarified by the authors before making a final assessment of the manuscript metabolites-2002846

Minor considerations :

1)     The manuscript is well written, however as minor comment I recommend doing a revision of the English language to avoid the repetition of terms to describe the function of proteins, cell pathways or their expression levels, such as "increased" or "decreased", and use / combine more as "triggers" or "promotes". However, I reiterate that the grammatical level of the text is adequate. However, I consider that some titles should be rewritten to make them more decisive in terms of the result and conclusion shown. For example: pages 220 and 232

2)     Important: the authors must reorganize the figures well. All the figures appear one after the other before beginning to explain the results, this makes reading tedious, having to continuously return to the beginning of the manuscript. Each section must be organized according to its figure. The final work of editing the manuscript needs organization, time and coherence.

3)     REWRITING: sentence 255-258, the explanation is not clear enough. Page 259-263, the authors should further break down the information in Figure 5. What conclusions do the authors of Figures 5A-5C draw? Title section 3.2, they must rewrite it to make it more conclusive and consistent with the work shown

Major considerations :

1)     The relationship of high blood cholesterol levels to tumor progression is unclear. The authors should delve into this relationship / explanation, providing bibliographic data in this regard. In case these measurements are only considered as positive controls of PA action, I recommend including it as supplementary material.

2)     The text between lines 220-231 is not clear enough. For example, between 226-229 what is the basis for looking at these markers? The authors must better link the experiments to give coherence to the manuscript and the results section. Moreover, discussion must be modified according these modifications.

3)     Section 3.2: Why TLR3 marker has been chosen?, the authors should better explain the data and how to analyze the Molecular Docking, controls and types of assay.

4)     Figure 6: the authors must corroborate the Peli1 siRNA data, comparing the effect of several independent siRNA sequences, even when dealing with a pool ... the sequences and their effects must be individualized.

5)     The authors should show as supplementary material the complete WBs, at least for the MMP2 marker, is it the pro-active or active and processed form? Same for phospho-NFkB (total NFkb?).IMPORTANT: all the markers have been analyzed in the same gel? only one housekeeping actin is included...in case of different gels, it must be specified.

6)     Figure 7A: Could the authors quantify the signal intensity for the markers TLR4, TRIF and p-NFkb? Comparing NC controls, siPeli1, PA and siPeli1+PA

Author Response

(The authors gave the same response as above.)

Round 2

Reviewer 2 Report

Authors did all suggested revisions, and the paper titled: “Palmitic acid promotes lung metastasis of melanoma via TLR4/TRIF-Peli1-NF-κB pathway” is suitable for publication in Metabolites after text checking (for example 3. .Results should be 3. Results, and 5. conclusion should be 5. Conclusion). 

Author Response

Dear Reviewer,

Thank you for your comments on our manuscript entitled “Palmitic acid promotes lung metastasis of melanoma via TLR4/TRIF-Peli1-NF-κB pathway” (Manuscript Number: 2002846). We appreciated your thoughtful review and have corrected these written errors in "3. Results" and "5. Conclusion". Revised sections are marked in red in the manuscript. 

Reviewer 3 Report

   Dear authors 

   Thank you very much for all your considerations, explanations and experiments

   Best Regards

   Reviewer

Author Response

Dear Reviewer,

Thank you for your thoughtful comments on our manuscript entitled “Palmitic acid promotes lung metastasis of melanoma via TLR4/TRIF-Peli1-NF-κB pathway” (Manuscript Number: 2002846). These comments are valuable and helpful for improving our manuscript, as well as the significantly guidance to our research.